# The Impact of CEO Educational Background on Corporate Risk-Taking in China

**Jinyi Zhang [1], Chunxiao Xue [1,2] and Jianing Zhang [1,2,*]**

1   College of Business and Public Management, Wenzhou-Kean University, Wenzhou 325060, China
2   Center for Big Data and Decision-Making Technologies, Wenzhou-Kean University, Wenzhou 325060, China
*   Correspondence: jianingz@wku.edu.cn

**Abstract:** This article investigates whether, and how, CEO educational background affects Chinese corporate risk-taking. Using a sample of 4681 firm-year observations from 2012 to 2020, we find that CEO educational background is negatively associated with corporate risk-taking. The nonlinear quadratic regression shows a convex relationship, consistent with the finding that the effect is more profound for the subsample with relatively lower education levels. The negative relationship is stronger for the firms with higher leverage, with lower tangibility, and in non-manufacturing industries. We also address the endogeneity issue using a two-stage least squares regression. This paper may provide valuable insights for shareholders, helping them to hire the most suitable CEOs to achieve shareholders' objectives and increase the corporation's competitiveness in the market.

**Keywords:** educational background; CEO; risk taking; Chinese stock market

**JEL Classification:** G11; G24; G32

## 1. Introduction

Why do some companies take on more risks than others? Since corporate risk-taking is essential to a company's long-term survival and growth (John et al. 2008; Baumol et al. 2007), understanding the problem can be helpful to the decision-making of shareholders and investors. Studies have emphasized strategic risks among different risk types, which refer to corporate strategic moves with various returns and unknown ruin (Baird and Thomas 1985), and other firm-level and country-level characteristics, such as board size, audit committee effectiveness, and the country's institutional quality (Wang 2012; Nguyen 2022a; Nguyen and Dang 2022). However, the focuses on strategic moves and other firm-level and country-level characteristics overlook the effects of managerial characteristics on corporate risk-taking.

This paper examines whether and how CEO educational background affects corporate risk-taking. The existing literature on traditional corporate governance assumes that managers are homogeneous. However, the upper echelon theory suggests that managerial characteristics play a role in corporation decisions (Hambrick 2007; Hambrick and Mason 1984). The financial crisis has revealed weaknesses in board structure, executive compensation, and the significance of risk management (Calomiris and Carlson 2016). Besides these impacting factors, numerous factors are combined to shape a corporation's attitude toward potential risk, including CEO demographic characteristics. According to Martino et al. (2020), the decision to perform a risky task can be interpreted as a result of the personality of the decision-makers. Indeed, CEOs predominate among executives, and therefore they dominate corporate decision makings. Thus, examining how a CEO—the most potent decision-maker (Minichilli et al. 2010)—affects a firm's risk-taking could provide insight into corporate risk-taking behaviors.

Our study fits in the literature that examines the effects of CEOs' educational backgrounds on corporate decision-making and policy outcomes. Given the significance of

risk-taking, some connect risk preferences with CEO education level. Although the literature on CEO personal traits has found that different educational backgrounds may lead to different viewpoints (Anderson et al. 2011), it remains unclear how CEO educational background is associated with corporate risk-taking. For example, some find that corporate risk-taking increases with CEOs' education level (Farag and Mallin 2018) because education gives CEOs the competence and confidence to deal with complex situations with unknown risks. In contrast, others find that CEOs with higher education levels are more risk averse (Martino et al. 2020), as a higher education level provides them with more opportunities and fewer incentives to take on additional risks.

To examine the relationship between CEOs' educational experience and corporate risk-taking, we employed a sample of 4681 firm-year observations of listed companies in China from 2012 to 2020. We found that CEO educational background and corporate risk-taking are negatively correlated. We used the variability of return on assets (ROA) over five years to measure corporate risk-taking. We adjusted ROA by the industry's economic cycle to have a cleaner measure of corporate risk-taking. The regression results show that CEO educational background negatively influences corporate risk-taking. In addition, the nonlinear quadratic regression shows a convex relationship. We then performed a subsample analysis. A significant impact of education on risk-taking exists in CEOs with a bachelor's degree or lower, while the impact of education loses significance for the subsample of CEOs with a master's degree or higher. It suggests that the relationship is more profound when CEOs' education level is lower, consistent with the results from quadratic regression.

Furthermore, we found that the negative impact of CEO educational backgrounds on corporate risk-taking is more profound when firms have higher leverage, lower tangibility, or belong to the non-manufacturing industry. We found an insignificant but negative relationship between education and risk-taking using an alternative risk-taking measure based on return on equity (ROE). Finally, using one- and two-year lagged education as instrumental variables, our two-stage least squares regression partially addresses the endogeneity concern.

Our study contributes to the literature examining the effects of CEO educational backgrounds on corporate decision-making and outcomes. Jaggia and Thosar (2021) used a sample of US firms from 2011 to 2014 to investigate the effects of CEOs' educational backgrounds on firms' outcomes. They find that CEOs with elite education are more likely to hold cash and invest in research and development (R&D). Using a sample of Indonesian firms, Harymawan et al. (2020) find a similar result that CEOs with higher education levels invest more in R&D. Martino et al. (2020) examine the relationship between CEO characteristics and risk-taking in a sample of Italian family firms and find that the company's risk-taking is negatively associated with the CEO education level. Cho et al. (2021) study a sample of listed Chinese manufacturing firms and find a moderating effect of executive education level on the relationship between gender diversity and bankruptcy risk. Farag and Mallin's (2018) study is closely related to our work. They investigate the influence of CEO demographic characteristics on corporate risk-taking for Chinese IPOs in the period of 1999–2009. Measuring CEO education by a dummy variable that equals one if the CEO holds a postgraduate degree and zero otherwise, they find that CEOs with a higher level of education are more likely to make risky decisions. However, their study focuses on the IPOs in China, while ours focuses on the listed companies in China in a more recent time period. Moreover, their measure of corporate risk-taking differs from ours. To summarize, the existing literature finds mixed results about the effects of CEO educational background on corporate risk-taking. Using a more generalized sample of listed companies in China rather than IPOs firms, firms in a specific industry, or family firms, we find that CEO educational background is negatively associated with corporate risk-taking. In contrast with previous studies, we also explore a nonlinear quadratic relationship and moderating effects of leverage, assets tangibility, and industry heterogeneity. This study provides valuable insights for shareholders, helping them to hire the most suitable CEOs

to achieve the preferred degree of risk-taking behaviors (Amihud and Lev 1981; Li and Roberts 2018). If a firm pursues a risk-mitigating strategy, then it is wise to choose a CEO whose education qualification is higher. On the other hand, a CEO with a lower education qualification tends to explore strategies with a broad risk spectrum.

The rest of the paper is as follows: Section 2 reviews the existing literature on this topic and hypothesizes. Section 3 describes the data and methodologies used in this paper. Section 4 investigates the relationship between CEO educational background and corporate risk-taking and performs various robustness checks. Section 5 concludes.

## 2. Literature Review and Hypotheses Development

The effects of CEO educational background on corporate risk-taking behaviors are theoretically complex and empirically ambiguous. We review relevant theories and empirical evidence in the following sections.

### 2.1. CEO Educational Background

The upper echelon theory posits that executives' experiences can significantly affect their choices (Hambrick and Mason 1984; Hambrick 2007). Therefore, we argue that CEOs, as the chiefs among all executives, make decisions based on their experiences. Moreover, educational background is one of the experiences that shape individuals' characteristics.

Researchers can apply different methods to evaluate a CEO's educational background. Attah-Boakye et al. (2021) use two variables to assess education level: academic and professional qualifications. In their study, education is regarded as a proxy for cognition, as education impacts cognitive biases.

### 2.2. Corporate Risk-Taking

Corporate risk-taking is a reflection of CEOs' choices. Understanding corporate risk-taking is essential because risks are critical to the survival and performance of firms. There is an extensive body of literature about how corporate risk-taking is affected by firm-level characteristics, such as the audit committee effectiveness, the presence of risk committee, corporate governance, and capital buffer policy (Bhuiyan et al. 2021; Nguyen 2022a; Dang and Nguyen 2021; Jiang et al. 2020). In addition, studies on country-level characteristics find that national culture, bank governance structure, and a country's institutional quality influence corporate risk-taking (Li et al. 2013; Nguyen 2022b; Nguyen and Dang 2022).

One risk-taking measure is the variability of the adjusted ROA. Faccio et al. (2011) calculate the standard deviation of the adjusted returns for each firm in the cross-sectional regressions during the entire sample period of 1999–2007. Similarly, Su et al. (2017) select five years as an observation phase and measure corporate risk-taking by calculating the standard deviation of the adjusted ROA. To obtain clearer insight into risk-taking, they remove the influence of the economic cycle.

Another approach is leverage. Faccio et al. (2016) consider leverage as a measure of risk-taking. Leverage is defined as the ratio of financial debt divided by the sum of long-term debt, short-term loans, and equity. Higher leverage is related to more riskiness in corporate financing choices.

### 2.3. The Effects of CEO Educational Background on Risk-Taking

CEO educational background is essential in corporate risk-taking decisions, as their prior academic experience biases their perceptions of new ideas. Orens and Reheul (2013) argue that CEOs' decisions reflect their education: highly educated CEOs tend to consider risky choices, accept innovative ideas, and are better informed about the external environment. This argument is supported by the cases that educated CEOs are likely to direct more innovative companies since better education levels allow them to deal with new information effectively (Ramón-Llorens et al. 2017). Lin et al. (2011) claim that CEOs with professional backgrounds are more inclined to take innovative strategies and more skillful in business decision-making. In addition, Barker and Mueller (2002) find a positive rela-

tionship between an advanced science degree and investment in R&D, as better-educated CEOs tend to have greater cognitive complexity to absorb new ideas. CEOs' education level is positively related to their information processing ability. By reducing the cost of information processing, CEOs are more likely to be open to new ideas and face less uncertainty (Black et al. 2018; Knight et al. 2003; Pressley et al. 1989). Following this line of arguments, education is positively associated with corporate risk-taking.

Some studies divide education levels into different divisions in detail. Farag and Mallin (2018) investigate a sample of 892 Chinese IPOs and find that CEOs with postgraduate qualifications are more likely to consider risky decisions. Similarly, Sitthipongpanich and Polsiri (2012) survey 1356 companies on the Stock Exchange of Thailand from 2001 to 2005. They show that highly educated CEOs are more confident and willing to take more risks. Moreover, Bertrand and Schoar (2003) find that CEOs with MBA degrees tend to follow more aggressive strategies. Beber and Fabbri (2012) study US non-financial firms by constructing panel data of foreign currency derivative holdings and currency exposures. They find that CEOs with MBA degrees are likely to engage in risk-taking because they might be overconfident. In addition, CEOs with advanced science degrees have better risk-taking skills and perform better in the same situation (Tyler and Steensma 1998).

However, others argue that CEO education level is negatively associated with risk-taking behavior because they know how to better avoid risks (Boubaker et al. 2020; Jung 2015; Martino et al. 2020). Using a natural experiment setting, Jung (2015) finds that more education decreases risk-taking because individuals become more aware of the risks they face. However, she also finds that the adverse effects of education level on risk-taking only exist among those with low education, indicating a nonlinear relationship between education and risk-taking. Likewise, Grable and Lytton (1998) argue that educational background is the critical factor of risk-taking intensity in businesses. Educated CEOs are likely to construct formalized procedures to minimize loss and have a more vital ability to acquire, absorb, and transform knowledge. They can follow existing systems to avoid risks rather than conduct new things with unknown dangers. The context of education also justifies it. During their experience of higher education, CEOs have learned how to diminish risks and chase long-term development (Orens and Reheul 2013).

Some empirical studies support that a company's risk-taking is negatively related to CEO educational background. For example, Martino et al. (2020) analyze a sample of 107 Italian family firms listed on the Milan Stock Exchange. They find that CEOs with higher education levels tend to be risk-avoiders. Boubaker et al. (2020) investigate the effects of French post-secondary educational institutions. They discover that hedging decisions, which are used to reduce risks, are more frequent when the education quality is higher. They argue that derivative usage enhances firm performance only when CEOs are from elite institutions. Iqbal (2015) and Cheng et al. (2010) also find similar results by examining hedging decisions in the oil and gas industries. In addition, Jaggia and Thosar (2021) find that CEOs with a science and technology background tend to increase R&D spending and decrease exposure to financial risk for the company.

Other researchers argue that education experience does not have a noticeable effect on a firm's risk-taking, which is indicated by R&D spending (Daellenbach et al. 1999; Barker and Mueller 2002), cash holdings (Bertrand and Schoar 2003; Mun et al. 2017), and entrepreneurial behaviors (Wang and Poutziouris 2010; Nilmawati et al. 2021). Wang and Poutziouris (2010) conduct a quantitative survey based on 236 companies. They find no significant relationship between CEO education level and entrepreneurial risk-taking. Nilmawati et al. (2021) also find a similar result using a sample of 283 non-financial companies on the Indonesia Stock Exchange from 2010 to 2017.

To summarize, the existing literature finds mixed evidence about the effects of CEO educational background on corporate risk-taking behaviors. This study aims to fill the gap by studying the impact of CEOs' educational background on corporate risk-taking behaviors in the context of Chinese entrepreneur firms. Based on the above discussions,

higher educated CEOs tend to reduce risks and chase long-term development (Orens and Reheul 2013). We propose the following hypotheses:

**Hypothesis 1.** *The CEO's education level is negatively related to the firm's risk-taking.*

**Hypothesis 2.** *The negative impact of the CEO's education level on the firm's risk-taking is more profound when the education level is lower.*

### 3. Data and Methodology

*3.1. Data*

The study employed the China Stock Market and Accounting Research (CSMAR) database, the most frequently used database on the Chinese capital market (Xiong et al. 2020). From it, we obtained CEO background information from December 1990 to September 2021. We also obtained financial indicators information from December 2003 to December 2020 for all the companies on the Shanghai and Shenzhen Stock Exchange markets. Disclosure requirements in China require publicly listed companies to disclose their annual accounting data. Therefore, these financial indicators are available each year. However, most CEOs' educational background information is missing before December 2012. Consequently, the sample period is from 31 December 2012 to 31 December 2020. We excluded the firms with incomplete datasets. The final sample consists of 4681 firm-year observations, representing 937 unique firms from 2012 to 2020.

As for CEO background information, CSMAR reports company security code, CEO's name, gender, age, graduate school, educational background, major, professional background, overseas experience, and academic background. Education is divided into seven categories: technical secondary school and below as one, college as two, bachelor as three, master as four, doctor as five, others as six, and MBA/EMBA as seven. Because a doctor is usually described as a more pre-eminent degree than an MBA, we recoded the MBA as five, the doctor as six, and removed a few observations of the others category. Figure 1 shows the time trend of each degree category in the CSMAR database from 2012 to 2020, where there is an apparently increasing trend for the master's degree. CSMAR also reports foreign work and study experiences. We added a dummy variable, which equals one if the CEO has the foreign experience and zero otherwise.

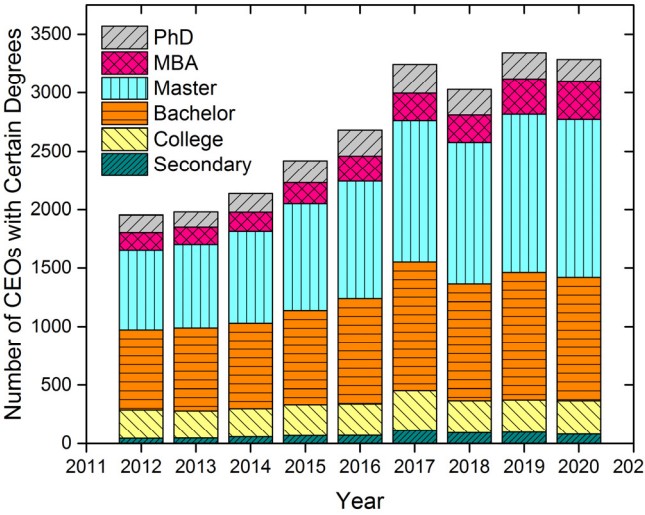

**Figure 1.** Time trend of the number of CEOs with certain educational degrees. The figure shows the number of CEOs with certain educational degrees from 2012 to 2020 using the CSMAR database.

As for the company's risk-taking, we obtained from the financial indicators of CSMAR, the company security code, ROA, the growth rate of operating income, asset-liability ratio, enterprise type, company size, and market value.

As for other control variables, we obtain from the balance sheet of CSMAR the firm size (the natural logarithm of total firm assets), leverage (the company's entire liability divided by total assets), and asset tangibility (the net fixed assets divided by the total assets).

From the ownership information form of CSMAR, we also collected the company age as another control variable. The establishment date of the company was subtracted from the observed date to obtain the number of months. If the rest of the days after subtraction is more than 15 days, then the number of months plus one and the sum should be divided by 12 to obtain the company's age; if not, the company's age is the number of months divided by 12. Finally, we took the logarithm of one plus company age to reduce the skewness.

### 3.2. Methodology

The educational background of CEOs could be measured by CEOs' academic backgrounds (Attah-Boakye et al. 2021). This study divided a CEO's educational background into six categories.

As for a firm's risk-taking, this paper used the variability of ROA to measure corporate risk-taking (John et al. 2008; Faccio et al. 2011; Faccio et al. 2016; Su et al. 2017). ROA is defined as the ratio of earnings before interests and taxes to the total assets. Before calculating the variability of ROA, it is necessary to remove the effect of industry's economic cycle to have a cleaner measure of the level of risk resulting from the corporate operating decision (Faccio et al. 2011). We used the following equation to calculate the industry-adjusted ROA:

$$AdjROA_{ijt} = ROA_{ijt} - \frac{1}{n_{jt}} \left( \sum_{k=1}^{n_{jt}} ROA_{ijt} \right) \tag{1}$$

where $AdjROA_{ijt}$ is the return on assets after adjustment, $ROA$ refers to return on assets, the subscript $i$, $j$, and $t$ denote company, industry, and year, respectively, $n_{jt}$ represents the number of companies in industry $j$ and year $t$.

After obtaining the adjusted ROA, we calculated the variability of ROA to measure corporate risk-taking. We selected five years as an overlapping observation phase ($T = 5$): 2012–2016, 2013–2017, 2014–2018, 2015–2019, and 2016–2020.

$$RT_{it} = Max\left(AdjROA_{ijt}, AdjROA_{ijt+1}, \ldots, AdjROA_{ijt+T}\right) - Min\left(AdjROA_{ijt}, AdjROA_{ijt+1}, \ldots, AdjROA_{ijt+T}\right) \tag{2}$$

where $RT$ represents the variability of ROA and $T$ represents an overlapping observation phase.

To investigate the relationship between CEO education and corporate risk-taking, we used panel-data regression with industry- and year-fixed effects (FE) as below.

$$RT_{it} = \begin{aligned} &\beta_0 + \beta_1 Education_{it} + \beta_2 Size_{it} + \beta_3 Leverage_{it} + \beta_4 Growth_{it} + \beta_5 Tangibility_{it} \\ &+ \beta_6 Age_{it} + \beta_7 Oversea_{it} + \text{Industry FE} + \text{Year FE} + \varepsilon_{it} \end{aligned} \tag{3}$$

where *Education* is the educational level of the CEO, *Size* is the logarithm of the total assets, *Leverage* is the total liability divided by the total assets, *Growth* is the growth rate of operating income (winsorized at the 1% and 99% levels to remove the impact of outliers), *Tangibility* is the net fixed assets divided by the total assets, *Age* is the logarithm of one plus the difference between the establishment date of the company and the observed year, and *Oversea* is a dummy variable that equals one if the CEO has a foreign study or work experience and zero otherwise.

Table 1 reports the sample statistics. The risk-taking measure has an average value of 0.12, a standard deviation of 0.37, and a skewness of 21.29. The risk-taking measure shows considerable variation and is strongly positively skewed. The average education is 3.67, which is primarily close to the master's and bachelor's degrees. The standard deviation of *Education* is 1.10. We did not observe extreme maximum values for most variables except for *Growth*. After being winsorized at the 99% level, *Growth* exhibits a moderate maximum value of 7.48.

**Table 1.** Summary Statistics.

| Variables | Obs. | Mean | StdDev | Min | Max | Skewness | Kurtosis |
|---|---|---|---|---|---|---|---|
| RT | 4681 | 0.1153 | 0.3668 | 0.0005 | 12.1303 | 21.2927 | 566.9021 |
| Education | 4681 | 3.6725 | 1.0988 | 1.0000 | 6.0000 | 0.2698 | 3.1735 |
| Size | 4681 | 22.1590 | 1.4699 | 16.7575 | 29.7596 | 1.4509 | 6.6718 |
| Leverage | 4681 | 0.4196 | 0.2978 | 0.0080 | 11.5097 | 15.2060 | 515.6679 |
| Growth | 4681 | 0.4071 | 0.9993 | −0.5918 | 7.4849 | 4.6739 | 30.1317 |
| Tangibility | 4681 | 0.1511 | 0.1437 | 0.0000 | 0.8758 | 1.2488 | 4.4721 |
| Age | 4681 | 15.5202 | 5.6444 | 3.3300 | 49.6700 | 0.4689 | 3.7762 |
| Oversea | 4681 | 0.1021 | 0.3028 | 0.0000 | 1.0000 | 2.6280 | 7.9066 |

The table reports descriptive statistics for the sample companies from 2012 to 2020.

Table 2 shows the correlations among variables. The risk-taking measure has significantly negative correlations with education and firm size. In contrast, risk-taking is positively correlated with financial leverage.

**Table 2.** Correlations.

| | RT | Education | Size | Leverage | Growth | Tangibility | Age | Oversea |
|---|---|---|---|---|---|---|---|---|
| RT | 1.0000 | | | | | | | |
| Education | −0.0646 ***<br>(0.0000) | 1.0000 | | | | | | |
| Size | −0.0799 ***<br>(0.0000) | 0.1697 ***<br>(0.0000) | 1.0000 | | | | | |
| Leverage | 0.2420 ***<br>(0.0000) | 0.0429 ***<br>(0.0034) | 0.3851 ***<br>(0.0000) | 1.0000 | | | | |
| Growth | 0.0184<br>(0.2084) | −0.0108<br>(0.4613) | −0.0059<br>(0.6864) | 0.0485 ***<br>(0.0009) | 1.0000 | | | |
| Tangibility | −0.0155<br>(0.2893) | −0.0672 ***<br>(0.0000) | −0.0763 ***<br>(0.0000) | −0.0322 **<br>(0.0277) | −0.1455 ***<br>(0.0000) | 1.0000 | | |
| Age | 0.0069<br>(0.6354) | 0.0208<br>(0.1557) | 0.1904 ***<br>(0.0000) | 0.1977 ***<br>(0.0000) | 0.0751 ***<br>(0.0000) | −0.0355 **<br>(0.0153) | 1.0000 | |
| Oversea | 0.0028<br>(0.8494) | 0.1461 ***<br>(0.0000) | −0.0352 **<br>(0.0161) | −0.0522 ***<br>(0.0004) | 0.0085<br>(0.5587) | −0.0214<br>(0.1428) | −0.0301 **<br>(0.0397) | 1.0000 |

The table shows the correlations among variables. *p*-values are reported below the estimated correlation coefficients in parentheses. ***, ** represent the statistical significance at the 1%, and 5% levels, respectively.

## 4. Results and Discussion

### 4.1. Baseline Regression

Our baseline regression is based on Equation (3), and the results are shown in Table 3. We added industry- and year-fixed effects to control for time-invariant characteristics for individual industries and years. We used five years as an overlapping observation phase. Columns 2 and 3 of Table 3 show negative coefficients on *Education* (−0.0215, −0.0182) at the 1% significance level after controlling for the industry- and year-fixed effects. The results suggest that CEOs with higher levels of education are more risk-averse, measured by the variability of ROA. Assuming a causal relationship, when a CEO obtains a higher level of degree, corporate risk-taking will decrease by 15.8% (= −0.0182/0.1153) of its mean. We also find that the coefficients of *Size* and *Age* are negative and statistically significant, indicating that larger and older firms are less risky. In addition, the significantly positive coefficient on *Leverage* suggests that higher-leverage firms are more likely to exhibit risk-taking. We do not find any significant effects on operating income growth, asset tangibility, or CEOs' overseas experience. Finally, the VIF values of the explanatory variables in Column 4 of Table 3 are all less than ten, indicating that multicollinearity is not an issue.

**Table 3.** Baseline regression results.

|  | *RT* | *RT* | **VIF Values** |
|---|---|---|---|
| *Education* | −0.0215 *** <br> (0.0049) | −0.0182 *** <br> (0.0048) | 1.16 |
| *Size* | | −0.0500 *** <br> (0.0047) | 1.99 |
| *Leverage* | | 0.4140 *** <br> (0.0188) | 1.32 |
| *Growth* | | −0.0011 <br> (0.0054) | 1.21 |
| *Tangibility* | | −0.0577 <br> (0.0396) | 1.36 |
| *Age* | | −0.0021 ** <br> (0.0010) | 1.31 |
| *Oversea* | | 0.0068 <br> (0.0168) | 1.08 |
| Constant | 0.2343 ** <br> (0.0930) | 1.1771 *** <br> (0.1319) | |
| Industry FE | Yes | Yes | |
| Year FE | Yes | Yes | |
| Observations | 4681 | 4681 | |
| $R^2$ | 0.094 | 0.185 | |

The table shows the panel regressions with industry- and year-fixed effects and corresponding VIF values for testing the multicollinearity issue. The standard errors are reported below the estimated coefficients in parentheses. ***, ** represent the statistical significance at the 1%, and 5% levels, respectively.

### 4.2. Nonlinear Quadratic Relationship

To investigate whether CEO educational background is nonlinearly related to the company's risk-taking, we add the squared term of *Education* as shown below.

$$RT_{it} = \begin{aligned}& \beta_0 + \beta_1 Education_{it} + \beta_2 Education_{it}^2 + \beta_3 Size_{it} + \beta_4 Leverage_{it} + \beta_5 Growth_{it} \\ & + \beta_6 Tangibility_{it} + \beta_7 Age_{it} + \beta_8 Oversea_{it} + \text{Industry FE} + \text{Year FE} + \varepsilon_{it}\end{aligned} \tag{4}$$

Our regression results of testing nonlinear quadratic relationship are shown in Table 4. In Columns 2 and 3 of Table 4, the coefficients on the squared term of *Education* (0.0127, 0.0141) are positive and significant at the 1% significance levels. Given the negative coefficients on *Education* (−0.1189, −0.1262) at the 1% significance level, the overall effects of educational background on corporate risk-taking are negative but at a decreasing rate. Substituting the mean education level (3.6725) into Equation (4), we find the economic contributions to the risk-taking are −0.46 (= −0.1262 × 3.6725) and 0.19 (= 0.0141 × 3.67252) for the education and the squared education terms, respectively. The effects of the control variables in Table 4 are similar to those in Table 3.

**Table 4.** Nonlinear quadratic relationship.

|  | *RT* | *RT* |
|---|---|---|
| *Education* | −0.1189 *** <br> (0.0239) | −0.1262 *** <br> (0.0227) |
| *Education*$^2$ | 0.0127 *** <br> (0.0031) | 0.0141 *** <br> (0.0029) |
| *Size* | | −0.0510 *** <br> (0.0047) |
| *Leverage* | | 0.4148 *** <br> (0.0188) |
| *Growth* | | −0.0005 <br> (0.0054) |
| *Tangibility* | | −0.0642 <br> (0.0396) |

**Table 4.** *Cont.*

|  | *RT* | *RT* |
|---|---|---|
| *Age* |  | −0.0021 ** |
|  |  | (0.0010) |
| *Oversea* |  | 0.0082 |
|  |  | (0.0167) |
| Constant | 0.4157 *** | 1.3966 *** |
|  | (0.1026) | (0.1391) |
| Industry FE | Yes | Yes |
| Year FE | Yes | Yes |
| Observations | 4681 | 4681 |
| $R^2$ | 0.097 | 0.189 |

The table shows regression results after adding the squared term of *Education*. The standard errors are reported below the estimated coefficients in parentheses. ***, ** represent the statistical significance at the 1%, and 5% levels, respectively.

### 4.3. Subsamples of Low and High Education Levels

We performed a robustness check on the nonlinear quadratic relationship. In Table 5, we split our sample into two subsamples based on CEO educational degrees: one subsample is for CEOs with a bachelor's degree or under, and the other is for CEOs with a degree higher than a bachelor's degree. We then ran separate regressions following Equation (3) for these two subsamples. Table 5 shows that the coefficients on *Education* are negative but only significant (−0.0841, −0.0868) in the subsample of CEOs with a bachelor's degree or under, suggesting that the effects of educational background only exist among CEOs without post-undergraduate degrees.

**Table 5.** Subsample analysis based on education levels.

|  | *RT* | *RT* | *RT* | *RT* |
|---|---|---|---|---|
| *Education* | Bachelor's degree or under | | Degree higher than a bachelor | |
|  | −0.0841 *** | −0.0868 *** | −0.0049 | −0.0015 |
|  | (0.0220) | (0.0202) | (0.0044) | (0.0045) |
| *Size* |  | −0.0769 *** |  | −0.0180 *** |
|  |  | (0.0108) |  | (0.0029) |
| *Leverage* |  | 0.6584 *** |  | 0.0466 *** |
|  |  | (0.0335) |  | (0.0142) |
| *Growth* |  | 0.0005 |  | 0.0024 |
|  |  | (0.0095) |  | (0.0043) |
| *Tangibility* |  | −0.1400 * |  | −0.0499 * |
|  |  | (0.0839) |  | (0.0258) |
| *Age* |  | −0.0036 * |  | 0.0003 |
|  |  | (0.0020) |  | (0.0007) |
| *Oversea* |  | 0.0189 |  | 0.0019 |
|  |  | (0.0465) |  | (0.0095) |
| Constant | 0.3878 * | 1.9354 *** | 0.1943 *** | 0.5395 *** |
|  | (0.1980) | (0.2906) | (0.0601) | (0.0854) |
| Industry FE | Yes | Yes | Yes | Yes |
| Year FE | Yes | Yes | Yes | Yes |
| Observations | 2062 | 2062 | 2619 | 2619 |
| $R^2$ | 0.123 | 0.270 | 0.149 | 0.163 |

The table shows regression results for two subsamples based on CEOs' education levels. The standard errors are reported below the estimated coefficients in parentheses. ***, * represent the statistical significance at the 1%, and 10% levels, respectively.

Assuming a causal relationship, when a CEO with a degree lower than a bachelor's degree obtains a higher degree, the company's risk-taking will be decreased by 75.3% (=−0.0868/0.1153) of its mean. The economic contribution of *Education* to the low-education subsample (75.3%) is five times the corresponding number for the full sample (15.8%) in Table 3. In addition, the coefficient on *Education* for the low-education subsample (−0.0868) is much larger in magnitude than the coefficient for the high-education subsample (−0.0015).

Overall, the results in Table 5 confirm the nonlinear quadratic relationship in Table 4, i.e., the effect of CEO educational background on corporate risk-taking is negative but at a decreasing rate.

### 4.4. Interaction Terms

To examine the moderating roles of specific firm-level characteristics on the effects of CEO education on corporate risk-taking, we performed further analysis by adopting two interaction terms: the interaction between *Education* and *Leverage*; the interaction between *Education* and *Tangibility*. We chose *Leverage* and *Tangibility* as the interaction terms because we hypothesized that these two variables are mostly related to corporate risk-taking behaviors. The higher the financial leverage, the riskier the firm becomes. Tangibility, the proportion of fixed assets, measures the firm's liquidity risk. We wanted to disentangle the effects of these two variables from the effect of education and therefore designed the regression equations below.

$$
\begin{aligned}
RT_{it} = \ & \beta_0 + \beta_1 Education_{it} + \beta_2 Education_{it} \times Leverage_{it} + \beta_3 Size_{it} + \beta_4 Leverage_{it} + \beta_5 Growth_{it} \\
& + \beta_6 Tangibility_{it} + \beta_7 Age_{it} + \beta_8 Oversea_{it} + \text{Industry FE} + \text{Year FE} + \varepsilon_{it}
\end{aligned}
\tag{5}
$$

$$
\begin{aligned}
RT_{it} = \ & \beta_0 + \beta_1 Education_{it} + \beta_2 Education_{it} \times Tangibility_{it} + \beta_3 Size_{it} + \beta_4 Leverage_{it} + \beta_5 Growth_{it} \\
& + \beta_6 Tangibility_{it} + \beta_7 Age_{it} + \beta_8 Oversea_{it} + \text{Industry FE} + \text{Year FE} + \varepsilon_{it}
\end{aligned}
\tag{6}
$$

The results are presented in Table 6. A significantly negative coefficient on the interaction term between *Education* and *Leverage* ($-0.1659$) at the 1% significance level indicates that CEOs in higher-leverage firms are more subject to the negative influence of their educational background in corporate risk-taking. On the other hand, we find a significantly positive coefficient on the interaction term between *Education* and *Tangibility* (0.0784) at the 5% significance level, suggesting that CEOs in firms with more tangible assets are less subject to the negative influence of their educational background on corporate risk-taking. Thus, both *Leverage* and *Tangibility* are moderators of the effects of CEO educational background on corporate risk-taking.

**Table 6.** The interaction between education and firm characteristics.

|  | *RT* | *RT* |
|---|---|---|
| *Education* | 0.0510 *** (0.0093) | −0.0306 *** (0.0069) |
| *Education × Leverage* | −0.1659 *** (0.0192) | |
| *Education × Tangibility* | | 0.0784 ** (0.0315) |
| *Size* | −0.0447 *** (0.0047) | −0.0508 *** (0.0047) |
| *Leverage* | 0.9794 *** (0.0681) | 0.4149 *** (0.0188) |
| *Growth* | 0.0003 (0.0053) | −0.0011 (0.0054) |
| *Tangibility* | −0.0593 (0.0393) | −0.3481 *** (0.1234) |
| *Age* | −0.0013 (0.0010) | −0.0021 ** (0.0010) |
| *Oversea* | −0.0004 (0.0167) | 0.0058 (0.0168) |
| Constant | 0.8246 *** (0.1371) | 1.2391 *** (0.1341) |
| Industry FE | Yes | Yes |
| Year FE | Yes | Yes |
| Observations | 4681 | 4681 |
| $R^2$ | 0.198 | 0.186 |

The table shows regression results after adding the interaction terms between education and leverage or asset tangibility. The standard errors are reported below the estimated coefficients in parentheses. ***, ** represent the statistical significance at the 1%, and 5% levels, respectively.

### 4.5. Manufacturing vs. Non-Manufacturing Industries

To test the industry heterogeneity, we divided our sample into different industries. There are 17 different industries in our sample. However, two-thirds of the sample are in the single manufacturing industry. In order to have a matched comparison, we divided our sample into two subsamples belonging to the manufacturing and non-manufacturing industries. We created a *ManufacturingDummy*, which equals one if the company is in the manufacturing industry and zero otherwise. We then added the interaction term between *ManufacturingDummy* and *Education* in the following equation.

$$
\begin{aligned}
RT_{it} = \ & \beta_0 + \beta_1 Education_{it} + \beta_2 Education_{it} \times ManufacturingDummy_{it} + \beta_3 Size_{it} + \beta_4 Leverage_{it} \\
& + \beta_5 Growth_{it} + \beta_6 Tangibility_{it} + \beta_7 Age_{it} + \beta_8 Oversea_{it} + \text{Industry FE} + \text{Year FE} + \varepsilon_{it}
\end{aligned}
\tag{7}
$$

The results are given in Column 2 of Table 7. The coefficient on the interaction term between *ManufacturingDummy* and *Education* (0.0180) is positive and significant at the 10% significance level, suggesting that the negative impact of CEOs' education on corporate risk-taking is mitigated for the manufacturing industry. Likewise, we split our sample into two subsamples in manufacturing and non-manufacturing industries and ran regressions following Equation (3). The coefficient on *Education* in the manufacturing industry ($-0.0133$) is less in magnitude than the corresponding coefficient in the non-manufacturing industry. Overall, the negative impact of CEO educational background on risk-taking exists for both industries but is stronger in non-manufacturing industries than in manufacturing industries.

**Table 7.** Manufacturing vs. non-manufacturing industries.

|  | *RT* | *RT* | *RT* |
|---|---|---|---|
|  |  | Manufacturing | Non-manufacturing |
| *Education* | −0.0317 *** | −0.0133 *** | −0.0344 *** |
|  | (0.0094) | (0.0049) | (0.0114) |
| *Education × ManufacturingDummy* | 0.0180 * |  |  |
|  | (0.0109) |  |  |
| *Size* | −0.0502 *** | −0.0569 *** | −0.0175 * |
|  | (0.0047) | (0.0052) | (0.0101) |
| *Leverage* | 0.4133 *** | 0.4626 *** | −0.0082 |
|  | (0.0188) | (0.0177) | (0.0692) |
| *Growth* | −0.0012 | 0.0091 | −0.0054 |
|  | (0.0054) | (0.0068) | (0.0090) |
| *Tangibility* | −0.0565 | −0.0721 * | −0.0288 |
|  | (0.0396) | (0.0411) | (0.0925) |
| *Age* | −0.0020 ** | −0.0021 ** | −0.0001 |
|  | (0.0010) | (0.0010) | (0.0024) |
| *Oversea* | 0.0056 | 0.0178 | −0.0365 |
|  | (0.0168) | (0.0175) | (0.0388) |
| Constant | 1.2278 *** | 1.2356 *** | 0.6539 *** |
|  | (0.1353) | (0.1175) | (0.2399) |
| Industry FE | Yes | Yes | Yes |
| Year FE | Yes | Yes | Yes |
| Observations | 4681 | 3209 | 1472 |
| $R^2$ | 0.186 | 0.239 | 0.144 |

The table shows the regression results after adding the interaction term between *Education* and *ManufacturingDummy* and for two subsamples in manufacturing and non-manufacturing industries. The standard errors are reported below the estimated coefficients in parentheses. ***, **, * represent the statistical significance at the 1%, 5%, and 10% levels, respectively.

### 4.6. Alternative Risk-Taking Measure

We employed an alternative measure of corporate risk-taking by using the variability of ROE. In Table 8, we find that the coefficient on *Education* is negative but insignificant. Still, the sign of the effect is consistent with our baseline results in Table 3.

**Table 8.** Alternative risk-taking measures using ROE.

|  | *RT2* | *RT2* |
| --- | --- | --- |
| *Education* | −0.1276 | −0.1100 |
|  | (0.1250) | (0.1282) |
| *Size* |  | 0.0678 |
|  |  | (0.1257) |
| *Leverage* |  | −0.3641 |
|  |  | (0.5051) |
| *Growth* |  | 0.1043 |
|  |  | (0.1442) |
| *Tangibility* |  | 0.0772 |
|  |  | (1.0644) |
| *Age* |  | 0.0069 |
|  |  | (0.0265) |
| *Oversea* |  | −0.5519 |
|  |  | (0.4506) |
| Constant | 1.0897 | −0.5557 |
|  | (2.3692) | (3.5414) |
| Industry FE | Yes | Yes |
| Year FE | Yes | Yes |
| Observations | 4681 | 4681 |
| $R^2$ | 0.059 | 0.060 |

The table shows the regression results with an alternative risk-taking measure using ROE. The standard errors are reported below the estimated coefficients in parentheses.

### 4.7. Two-Stage Least Squares

We addressed the endogeneity issue by running a two-stage least squares regression. Given the challenges of finding a valid instrumental variable, we followed Gregorio and Lee (2002) and Biyase and Zwane (2015), who used the lagged value of education as an instrument in their studies. We employed the education levels during the past two years as our instrumental variables. The regression equations are given below.

$$
\begin{aligned}
Education_{it} = \ & \beta_0 + \beta_1 Education_{it-1} + \beta_1 Education_{it-2} + \beta_3 Size_{it} + \beta_4 Leverage_{it} + \beta_5 Growth_{it} \\
& + \beta_6 Tangibility_{it} + \beta_7 Age_{it} + \beta_8 Oversea_{it} + \text{Industry FE} + \text{Year FE} + \varepsilon_{it}
\end{aligned} \tag{8}
$$

$$
\begin{aligned}
RT_{it} = \ & \beta_0 + \beta_1 \widehat{Education}_{it} + \beta_2 Size_{it} + \beta_3 Leverage_{it} + \beta_4 Growth_{it} + \beta_5 Tangibility_{it} \\
& + \beta_6 Age_{it} + \beta_7 Oversea_{it} + \text{Industry FE} + \text{Year FE} + \varepsilon_{it}
\end{aligned} \tag{9}
$$

The results are given in Table 9. In the first stage, both the one- and two-year-lagged education levels are significant predictors and thus strong instrumental variables for the current education level. In the second stage, the coefficient on the predicted education level is negative and statistically significant at the 5% level, suggesting that endogeneity may not be a severe issue in the current setting.

**Table 9.** Two-stage least squares.

|  | **First Stage** | **Second Stage** |
| --- | --- | --- |
|  | *Education* | *RT* |
| *Predicted Education* |  | −0.0158 ** |
|  |  | (0.0077) |
| *1-year Lagged Education* | 0.7415 *** |  |
|  | (0.0199) |  |
| *2-year Lagged Education* | 0.0890 *** |  |
|  | (0.0196) |  |
| *Size* | 0.0040 | −0.0300 *** |
|  | (0.0113) | (0.0067) |

**Table 9.** *Cont.*

|  | First Stage | Second Stage |
|---|---|---|
|  | *Education* | *RT* |
| *Leverage* | 0.1943 *** | 0.1153 *** |
|  | (0.0693) | (0.0409) |
| *Growth* | −0.0082 | 0.0101 |
|  | (0.0120) | (0.0071) |
| *Tangibility* | −0.1185 | −0.0547 |
|  | (0.0899) | (0.0530) |
| *Age* | 0.0013 | −0.0016 |
|  | (0.0022) | (0.0013) |
| *Oversea* | 0.1470 *** | −0.0061 |
|  | (0.0368) | (0.0219) |
| Constant | 0.5853 * | 0.8980 *** |
|  | (0.3061) | (0.1829) |
| Industry FE | Yes | Yes |
| Year FE | Yes | Yes |
| Observations | 2806 | 2806 |
| $R^2$ | 0.724 | 0.150 |

The table shows the two-stage least squares regression results using one- and two-year lagged education levels as instrumental variables. The standard errors are reported below the estimated coefficients in parentheses. ***, **, * represent the statistical significance at the 1%, 5%, and 10% levels, respectively.

## 5. Conclusions

Our study investigates the relationship between CEO educational background and corporate risk-taking. Our results suggest that CEO educational background is negatively associated with corporate risk-taking. The higher the CEO education level, the more likely the company is to avoid risks.

This paper has several contributions. First, our results contribute to the literature on the relationship between CEO educational background and corporate risk-taking in Chinese listed firms. We show that after controlling for various factors such as firm size, leverage, and asset tangibility, the CEO educational background has a significantly negative impact on corporate risk-taking. If the CEO educational background increases by one level, the risk-taking measure decreases by 15.8% of its mean. Second, there is a nonlinear convex relationship between CEO education and corporate risk-taking. This is consistent with the subsample analysis, where one level increase in the CEO's degree leads to a 75.3% reduction of the sample mean risk level, i.e., the main result is driven by the subsample of CEOs with a bachelor's degree or lower. Third, both high leverage and low tangibility play a strengthening moderating role in the negative relationship between CEO educational background and corporate risk-taking. Fourth, the negative impact of education on risk-taking is more prominent in the non-manufacturing industries than in the manufacturing industry, although the effects exist for both industries. Fifth, an alternative measure of risk-taking using the variability of ROE generates consistent but insignificant results. Finally, a two-stage least squares regression using lagged education levels as instrumental variables partially relieves the concern about the endogeneity issue.

This paper has several practical implications. As corporate risk-taking is essential for long-term survival and growth, this study can help investors judge corporate profiles. In addition, this paper offers shareholders some insights about the attribution of CEOs with different academic qualifications, thereby helping shareholders select suitable CEOs.

Our study has several limitations. First, the sample is limited to the Chinese listed firms and a short period of 2012–2020 due to data availability. Therefore, the generalization of the results should be made carefully. Second, we did not investigate the detailed CEOs' characteristics, such as the institution that offered the degree and the major of the degree, which may affect corporate risk-taking. We measure the educational background based on the degree level on an ordinal scale. Future research may consider the quality and the nature of the degree. Third, the available data only contain information about the CEO

who was in office at the end of the year. As a result, we did not focus on the possible effects of CEO turnover during the year. Fourth, we employ the variability of ROA as a measure of corporate risk-taking. Future research might employ alternative risk-taking measures, such as the change in leverage and variability of other performance measures. Finally, the dynamics of how the change in education level affects the change in risk-taking behavior might be explored in future research when a longer sample window and more detailed measures become available.

**Author Contributions:** Conceptualization, J.Z. (Jinyi Zhang); methodology, J.Z. (Jinyi Zhang) and J.Z. (Jianing Zhang); software, J.Z. (Jinyi Zhang) and J.Z. (Jianing Zhang); validation, C.X. and J.Z. (Jianing Zhang); formal analysis, J.Z. (Jinyi Zhang), C.X. and J.Z. (Jianing Zhang); investigation, C.X. and J.Z. (Jianing Zhang); resources, J.Z. (Jinyi Zhang); data curation, J.Z. (Jinyi Zhang); writing—original draft preparation, J.Z. (Jinyi Zhang); writing—review and editing, C.X. and J.Z. (Jianing Zhang); visualization, J.Z. (Jianing Zhang); supervision, J.Z. (Jianing Zhang); project administration, J.Z. (Jianing Zhang); funding acquisition, C.X. and J.Z. (Jianing Zhang) All authors have read and agreed to the published version of the manuscript.

**Funding:** This research was funded by the National Natural Science Foundation of China [No. 7197 2123], the Student Partnering with Faculty Research Program of Wenzhou-Kean University [No. WKU SPF202222], and the Internal Research Support Program of Wenzhou-Kean University [No. IR-SPG202205].

**Data Availability Statement:** The authors used China Stock Market and Accounting Research (CSMAR) data, which are publicly available through the CSMAR database.

**Conflicts of Interest:** The authors have no potential conflict of interest to declare.

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
