# Peer review of "The Impact of CEO Educational Background on Corporate Risk-Taking in China"

_jrfm, doi:10.3390/jrfm16010009_

Round 1

Reviewer 1 Report

Article:  The Impact of CEO’s Education Background on Corporate  Risk-taking in China

After reviewing this article, I think it is potential for publication but the authors should revise as comments below:

- In the introduction, the authors should make clear the research gap, I see some previous studies have the same topic, and the author should review them and emphasize your contribution. You said that “this paper contributed to the ongoing debate on the links between CEO 65 characteristics and firms’ risk behaviors”. It is not convincing when it does not specify what previous studies have not done.

Hypothesis development is not appropriate. Authors should not make opposing hypotheses. Although theories or empirical evidence make different predictions, the authors need to put the research in the right context and come up with a hypothesis that is most relevant. Especially, the authors must review the theories supporting these hypotheses. 

- In the hypothesis development section, the authors should update the recent studies. I find many previous studies relating to this topic but not to be reviewed in this study. So, I suggest that the authors review and cite the previous studies as follows: Jiang et al. (2020); Dang and Nguyen (2021); Nguyen (2022a); Nguyen and Dang (2022); Bhuiyan et al. (2020); Nguyen (2022b); …

- The value of the growth variable is very high, the author must scale it before using it for estimation. 

-          The authors need to add VIF values for the variables

-          In section 4, the authors must analyze the research results more deeply. I found that the authors did a rough analysis of the results, mostly reading the results from the results tables. The authors must be able to interpret the economic significance of the results in relation to the research context.

- The authors should show the limitation of this research in the conclusion.

Bhuiyan, M. B. U., Cheema, M. A., & Man, Y. (2020). Risk committee, corporate risk-taking and firm value. Managerial Finance, 47(3), 285-309. 

Dang, V. C., & Nguyen, Q. K. (2021). Internal corporate governance and stock price crash risk: evidence from Vietnam. Journal of Sustainable Finance & Investment, 1-18. doi:10.1080/20430795.2021.2006128

Jiang, H., Zhang, J., & Sun, C. (2020). How does capital buffer affect bank risk-taking? New evidence from China using quantile regression. China Economic Review, 60, 101300. 

Nguyen, Q. K. (2022a). Audit committee effectiveness, bank efficiency and risk-taking: Evidence in ASEAN countries. Cogent Business & Management, 9(1), 2080622. 

Nguyen, Q. K. (2022b). Determinants of bank risk governance structure: A cross-country analysis. Research in International Business and Finance, 60, 101575. doi:https://doi.org/10.1016/j.ribaf.2021.101575

Nguyen, Q. K., & Dang, V. C. (2022). Does the country’s institutional quality enhance the role of risk governance in preventing bank risk? Applied Economics Letters, 1-4. 

Reviewer 2 Report

This article investigates whether and how CEO's education experience affects Chinese corporate risk-taking. The paper is overall well written, and I only have two minor comments for the authors to consider.
1.    The current finding may be enhanced by considering some discussions on the differences of industries of firms.
2.    It may be better to include some figures to visualize temporal patterns of the data analyzed in this paper.

Round 2

Reviewer 1 Report

I am satisfied with this version. Congratulation.

Author Response

Thank you for your valuable comments, which greatly improve our manuscript.